# Spatial Uncertainty of Target Patterns Generated by Different Prediction Models of Landslide Susceptibility

**Andrea G. Fabbri** [1,*] and **Antonio Patera** [2]

1   DISAT, Università di Milano-Bicocca, 20126 Milan, Italy
2   Istituto Nazionale di Geofisica e Vulcanologia, 00143 Rome, Italy; antonio.patera@ingv.it
*   Correspondence: andrea.fabbri@unimib.it; Tel.: +39-3338313289

**Abstract:** This contribution exposes the relative uncertainties associated with prediction patterns of landslide susceptibility. The patterns are based on relationships between direct and indirect spatial evidence of landslide occurrences. In a spatial database constructed for the modeling, direct evidence is the presence of landslide trigger areas, while indirect evidence is the presence of corresponding multivariate context in the form of digital maps. Five mathematical modeling functions are applied to capture and integrate evidence, indirect and direct, for separating landslide-presence areas from the areas of landslide assumed absence. Empirical likelihood ratios are used first to represent the spatial relationships. These are then combined by the models into prediction scores, ordered, equal-area ranked, displayed, and synthesized as prediction-rate curves. A critical task is assessing how uncertainty levels vary across the different prediction patterns, i.e., the modeling results visualized as fixed, colored groups of ranks. This is obtained by a strategy of iterative cross validation that uses only part of the direct evidence to model the pattern and the rest to validate it as a predictor. The conducted experiments in a mountainous area in northern Italy point at a research challenge that can now be confronted with relative rank-based statistics and iterative cross-validation processes. The uncertainty properties of prediction patterns are mostly unknown nevertheless they are critical for interpreting and justifying prediction results.

**Keywords:** landslide susceptibility; ranking; cross validation; prediction model; prediction pattern; target pattern; uncertainty pattern

## 1. Introduction

Predicting landslide susceptibility of a region or study area has become a critical necessity with the continuing expansion of urbanizations across hazardous landscapes, increasing soil deterioration and the extensive damages inflicted by landslides [1]. Regional approaches to quantitative predictions have developed following the availability of thematic maps in digital form, including that of interpreted aerial photography and remotely sensed images. Examples of themes related to mass movement processes are expressions of various aspects of the rocks, soil, land cover, land use, soil permeability, groundwater table, and topographic surfaces [2,3].

In any application, the themes are hopefully representing the gravity-induced physical context of the process of slope failure. For this reason, the landslides are considered spatially related with the thematic map units or values over the study area of concern. These spatial relationships are established and integrated by mathematical models that transform raw data from specific landslide distribution maps and contextual maps into prediction maps or maps of the likelihood of future landslide occurrence [4].

Obviously, the detail and accuracy of the digital maps used for modeling must be congruous and the statistical relationships significant to obtain convincing likelihood maps of landslide occurrences that, hopefully, are more informative than what is already known. The term "convincing" must reflect the interpretable quality of the integrated likelihood

maps as spatiotemporal predictions and the certainty—or conversely, the uncertainty—associated with that quality.

The applications discussed in this contribution have the purpose of exposing aspects of spatial predictions that are commonly ignored—validation, robustness, and uncertainty of the resulting susceptibility levels expressed as relative ranks. They are the modeled values that make up the prediction patterns.

Analyzed are a database of landslide occurrences and their spatial context in a study area in northern Italy. The database has been the focus of experiments on comparison of models and analysis of modeling structure. Here, we will focus exclusively on the uncertainties affecting the ranks representing relative predicted levels of susceptibility to active landslides.

The following section deals with the predictive methodology, the terminology, and the analytical procedures applied. The next section describes the study area, its database for modeling, and previous research on landslide susceptibility. Experimental results follow on uncertainty identification and its relative measures. They consist of modeling prediction, target and uncertainty patterns, their interpretation, and comparison. Conclusions deem that uncertainty assessments are a desirable and necessary endeavor for making spatial prediction modeling a worthwhile practice.

## 2. Predictive Methodology, Terminology, and Analytical Procedures

A unified framework for modeling prediction patterns was proposed by Chung and Fabbri (1993) [5]. It was termed the "favorability function" and is used in the applications that are the focus of this contribution. The term favorability was selected for its generality in the meaning intended to be comprehensive and unified to cover a variety of mathematical interpretations. The following is a summary of the concepts in the modeling, the terminology used, and the processing strategy that enables us to assess spatial uncertainty.

A study area (SA) is first assumed to be selected by experts as sufficiently representative of the processes that are considered hazardous within it. Hopefully then, the statistics from spatial quantitative data available in the SA are suitable for information extraction by modeling. A second assumption is that we can have at least two subareas. First, an occurrence subarea is identified as affected by the hazardous landslide process, with the known presence of landslide trigger areas of a specific and congruous dynamic type. Then, another subarea is selected of non-occurrences in which landslides are known to be absent (or are unknown if present). The occurrence subarea is where the occurrences are located and characterized, and it is considered a training area (TA) for establishing the spatial relationships. Generally, the nonoccurrence subarea is the complement of the occurrence subarea within the SA. In some cases, however, the nonoccurrence subarea is only part of the SA or even outside the SA. This could also be the case with the occurrence area. Commonly, the occurrence subarea is a very small portion of the SA. A third assumption is that future occurrences within the SA will be similar in type and setting to the known ones.

To allow favorability function prediction modeling, a proposition is constructed as follows:

$F_i$: that a point *i* in the SA will be affected by a future landslide | the presence of spatial evidence. (1)

The symbol | indicates "given." The proposition is to be supported as true by means of the known occurrence distribution and their setting. The setting is considered spatial evidence. Therefore, the proposition links the occurrence locations (or their neighborhoods) and their setting as multivariate context. To satisfy the proposition as a mathematical statement, both the digital images representing the occurrence locations and their setting have to be transformed into spatial relationships. These new images have been termed the direct supporting pattern (DSP). They are the indexed occurrence locations. Indirect supporting patterns of the proposition (ISPs) are termed the images of their setting. The term "pattern" is used to indicate the results of this functional transformation into spatial relationships.

The landslide occurrences are converted into clusters of adjacent picture elements or pixels with a numerical value sequentially identifying each occurrence. The image

pixel values of the settings are transformed into normalized frequencies if derived from categorical maps such as lithology or land use. Continuous field digital images, such as elevation or slope angles, are transformed into density functions. The ratios of the normalized frequencies and the density functions for the TA are divided by those of the nonoccurrence subareas within the SA. These ratios are termed empirical likelihood ratios (ELRs). We could also apply other types of normalization; nevertheless, the end results would have the same purpose and functionality. ELRs, as described, range in value between zero and infinity and provide a measure of support of the proposition in (1).

Many different mathematical models can be used to convert and integrate the ratios into prediction patterns that classify the SA into levels of likelihood of future landslide occurrence. We will be using five different models in our applications but the favorability function modeling and the processing structure are independent of the mathematical models that will be indicated later on.

In essence, the likelihood ratios enable us to separate the presence of occurrence from their supposed absence within the SA. This separation uses another assumption. While the normalized frequencies or the density functions in the TA (the occurrence subarea) are related to observed/mapped landslide occurrences, the ones in the nonoccurrence subarea relate to both the absence of occurrences and the unknown occurrences. For the latter, we assume that their setting, due to being relatively rare, is diluted within the setting of the absence of occurrences. This assumption allows considering the ratios as a form of contrast between areas with presence and areas with a presumed absence of occurrences. Commonly, the TA with respect to the SA is three or four orders of magnitude smaller.

A favorability function at each point of a SA should have two properties for modeling, i.e.: (1) should be able to measure a relative level of likelihood that a point $i$ in the SA contains part of a future landslide of the given dynamic type and (2) should be able to provide a measure of uncertainty associated with the function by using only the part of all possible landslide occurrences present in the training area, TA.

Now suppose that we use a given prediction model, still unspecified at this point, that generates integrated scores as predicted values for every pixel of the SA. The scores are in some a-dimensional units or values between a minimum and a maximum. How do we establish the very high, high, intermediate, low, and very low likelihood of future landslide occurrence? This is not a simple question to answer, and it leads to further assumptions depending on the kind of data available for the SA, database of DSP and ISPs, and additional information.

Suppose, for instance, that we have temporal characterization for the many occurrences of the given specific dynamic type in the TA. We could use the occurrences from an older time interval to model a prediction pattern and then overlay it with the locations of the occurrences from the younger time interval. In this way, we obtain some statistics about their proportions within higher likelihood scores. Should some of them fall within the lower likelihood score areas, we would consider them as poorly predicted. Vice versa, well-predicted occurrences should fall on the higher likelihood score areas. We have termed this exercise as cross validation. In practice, this is a natural way to establishing how "good" our prediction pattern is as a "predictor of future occurrences."

Now let us suppose that the information on time partitioning of the occurrences, the DSP, is not available, as it is in most cases. How can we proceed with some other forms of cross validation? We could empirically pretend, for instance, not to know the existence of some of the occurrences, e.g., 25% of them, and use the remaining 75% to obtain a prediction pattern and then cross-validate it with the 25% we pretended not to know and that was not used as DSP for modeling. In this case, the "future" landslide occurrences for cross validation are the "next" 25%. Furthermore, we could operationally devise iterative strategies, depending on the number of occurrences available as DSP, such as (1) sequential exclusion of an arbitrary number of occurrences to be used for cross validation of the pattern produced with the remaining ones, (2) sequential selection of a number of occurrences for modeling and using the remainder for cross validation, or

(3) random selection of a number of occurrences repeated a convenient number of times. All these strategies will provide different ways to predicting the next arbitrarily selected number or proportion of occurrences. Recall that we have not yet selected any particular mathematical model for predicting.

Cross validation is a strategy for assessing the quality of our prediction modeling and also for comparing different prediction patterns, produced either by varying the number of ISPs or the mathematical models. The results of cross validations are tables of prediction scores for numbers or proportions of occurrence pixels in the SA. How do we interpret these dimensionless integrated scores generated for each pixel of a SA and ranging between a minimum and a maximum?

Given the number of transformations from the original map unit names or continuous values used to compute the ISPs and their conversion into relative scores, these are considered here as impossible to interpret directly by recognizing systematic changes or breaks. Instead, the prediction scores are easily converted into equal-area ranks after sequencing them in decreasing order. It was found convenient to obtain 200 ranks each corresponding to 0.5% of the SA. By equal-area ranking the prediction scores, it becomes practical and simple to display the prediction patterns and generate cross-validation tables of proportions of validation occurrences for each rank and cumulative tables of these proportions to be represented as curves.

A consequence of iterative cross validation is that a different prediction pattern is generated each time so that the set of patterns can be used to further characterize the initial prediction pattern obtained using all the occurrences in the DSP. We can then pile up the patterns and, if their numbers are sufficiently large, apply some form of statistics to obtain average and variance for each stuck of pixels in the SA. We have defined "target pattern" as what we wish to have—a representation of all past and future landslide occurrences (as susceptibility scores and associated uncertainty scores). In our case, some form of averaging the ranks of the prediction patterns from the iterations. We have defined "uncertainty patterns" as the expression of deviations from these averages. From practice, we have found that a very robust means for generating a target pattern is selecting the median rank for each pixel from the iteration prediction patterns, and for the uncertainty pattern, the rank of the ranges of deviations from the median ranks.

Furthermore, three assumptions must be reasonably satisfied [6], namely, (a1) the known landslide occurrences, the DSPs, are a "random selection" of all existing ones, known and unknown (allowing to extend the favorability function from the TA to the entire SA); (a2) the ISPs are correlated with the target pattern (allowing to estimate the function using the known part of the target pattern in the TA); and (a3) the process of slope failure is not random and follows a certain rule (allowing to model the favorability function).

Target and uncertainty patterns have opened the way to further characterize prediction patterns. Some of these will be discussed in Section 4. Let us now consider some favorability modeling functions of empirical likelihood ratios commonly applied and used here in our analyses. They are fuzzy set membership function [7], linear and logistic regression functions [8,9], empirical likelihood ratio function [10–12], and Bayesian predictive discriminant function [13]. We will abbreviate them as FZ, LI, LO, LR, and BP, respectively. The modeling functions imply different representation and combination rules of spatial relationships [5,7,14].

We will not discuss theoretical aspects of the modeling functions here because they were amply dealt with in the above-mentioned contributions [5,7,10–14]. The focus of our study is the characterization and interpretation of prediction patterns, independently of the particular prediction models applied. Our concern is that their applications to the same input data generate prediction patterns whose scores are in entirely different units, and these are considered as not interpretable or comparable except in terms of equal-area ranking. Moreover, they require entirely different combination rules. They imply diverse assumed interpretations of the spatial relationships expressed by the empirical likelihood ratios, and each interpretation combines ISPs with assumptions of conditional indepen-

dence (between categorical, continuous field ISPs and for integration of the two types) [12]. In geomorphology, and geosciences in general, maps are often spatially correlated so that such independence seldom exists or can be hypothesized. Nevertheless, it has been found that the existence of a correlation between ISPs causes minor alterations to equal-area rankings, mostly to the lower ranks [12,15].

### 3. Study Area, Database, and Previous Research

The Tirano South study area, whose location is shown in Figure 1, occupies the southern half of a community area termed "Comunità Montana Valtellina di Tirano" in the Province of Sondrio, Lombardy Region, in northern Italy. It was established in 1971 for socioeconomic and environmental protection (www.cmtirano.so.it; accessed on 7 April 2021). The geomorphology and landslide processes in the area were described by Sangalli (2008) [16], who compiled the available information and constructed a database for an initial landslide susceptibility analysis. Part of the data is being used in this contribution that also covers the same study area.

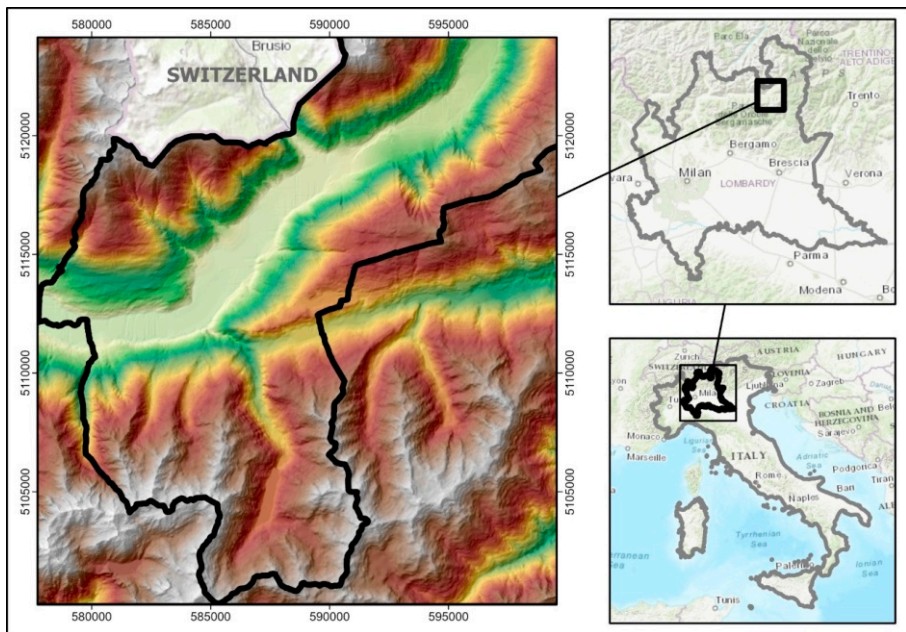

**Figure 1.** Locations of the Tirano South study area, in the Lombardy Region, northern Italy.

Elevations in this young Alpine geology area range between 300 and 3000 m a.s.l. and rainfall from 700 to 1900 mm, thus providing different climatic conditions. Vegetation consists of broad-leaved forests at lower elevations and coniferous forests at higher elevations. The geomorphology is controlled mostly by glacier activity, but anthropic activity significantly affects land use at valley bottoms with tourism, agriculture, and industries. Glacial and torrent erosion, together with intense rainfalls, are the triggers of slope instabilities. Out of a variety of landslide phenomena in the area, 70 active landslides were identified from published inventories. However, they did not contain sufficient information for separating rotational from translational dynamic types. In addition to the inventories used to generate digital images of landslide trigger area locations, various cartographies related to slope instability were available for compilation and digitization into the database, as shown in Figure 2. The database offers opportunities for experimenting on spatial prediction modeling of landslide susceptibilities, hazards, and risks.

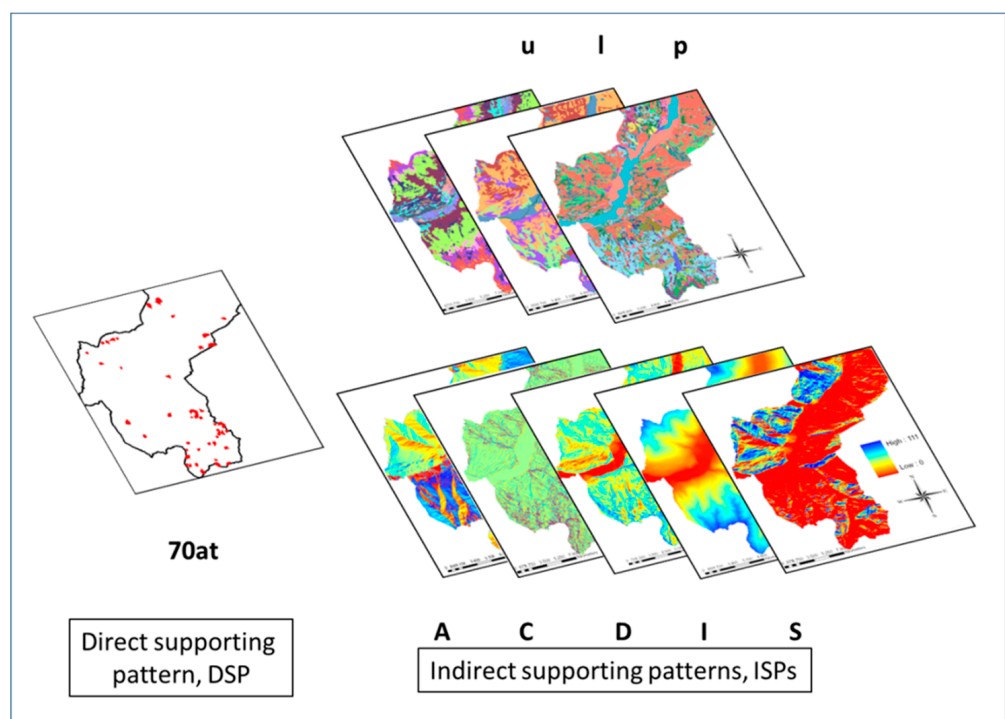

**Figure 2.** The Tirano South database is shown that will be converted into a direct supporting pattern (DSP) the 70 active landslide trigger areas, **70at,** and the three categorical and five continuous field digital maps to be converted into indirect supporting patterns (ISPs), **ulp** and **ACDIS.** The explanation is in the text.

The Tirano South database consists of digital images within a raster of 1090 pixels by 1194 lines with 20 m resolution. The study area proper within the raster covers 646,091 pixels corresponding approximately to 258 km$^2$. The trigger zones of the 70 active landslides, converted into DSP and abbreviated as **70at,** cover 697 pixels, i.e., as a training area (TA) a little less than 0.11% of the study area (SA). Eight cartographies are used to generate, at the same 20 m resolution, the ISPs consisting of maps with 23 land use classes, $\mathbf{u}_{1-23}$, with 51 geologic (lithologic) units, $\mathbf{l}_{1-51}$, and with eight permeability classes, $\mathbf{p}_{1-8}$ (three categorical maps), in addition to aspect (0° to 359°), **A**, topographic curvature (−32 to +29), **C**, topographic digital elevation (350–2906 m), **D**, internal relief (0–111 m for 3 × 3 pixels), **I**, and slope (0° to 61°), **S**, all derived from a 5 × 5 m resolution digital elevation model, DTM, resampled to 20 × 20 m (5 continuous field maps).

Previous research by Poli and Sterlacchini (2007) [17] on landslide susceptibility in the Tirano area used an earlier and simplified database with 28 active complex landslides and five binary and binarized factor maps applying the weights-of-evidence mathematical model and generating arbitrary thresholds of prediction ranks. That and similar applications of spatial prediction modeling were criticized [18] so that in joint contributions that used the same database, kindly shared by those authors, cross-validation procedures were preferred. The purpose was to assess the relative quality of the modeling results with an alternative model and different analytical processes that exposed the uncertainties associated with the prediction patterns [19,20]. A further study with a wealth of controlled information on Alpine landslides and three national–regional mapping initiatives provided new data for the construction of a high-quality database for the Tirano study area. It contained landslide inventories in addition to geological and soil–land use cartographies. For the northern part of the Tirano study area, prediction patterns were thus generated using both active and quiescent translational–rotational landslide scarps, the empirical likelihood ratio function model, and iterative cross-validation strategies based on blind tests [21].

Indeed, the Tirano study area had become the focus of much more research on landslide susceptibility modeling and other types of landslides and associated floods due to the wealth of information available and certified. Blahut et al. (2010a) [22] used two sets of aerial photographic coverages, 20 years apart, to select natural-condition debris flows for the estimation of the spatiotemporal probability of hazard initiation. The resulting distribution map was then used to model runout zone limitations using an empirical GIS-based simulation tool. The experiments indicated that the spatial variability observed needed validation tests for satisfactory interpretation. Furthermore, another study was carried out by those authors [23] of the degree of spatial pattern agreement between different landslide susceptibility maps that showed similar debris-flow predictive power. Their database contained the distribution of 573 scarps, and one half was used for modeling a susceptibility map, while the other half was used to validate the map. The similarity of predictive power, however, did not necessarily produce similar spatial configurations of susceptibility ranks, a fact that pointed at the variability of ranks and related spatial uncertainty.

Two more activities in the Tirano study area make it an exemplary instance of supporting background for natural hazard and risk studies. Blahut et al. (2010b) [24] constructed a reliable operational inventory of landslides from three available official inventories. Their tasks were (1) to prepare debris flow and factor maps inventories for susceptibility modeling, (2) generate indices of accountability/reliability and selection of factor maps, (3) evaluate/validate susceptibility maps, and (4) compare the results of different maps for combining them into an integrated susceptibility representation. Various methods of subdivision of debris flows and factor maps were obtained, including the separation of the Tirano study area into three more congruous subareas—northern, central, and southern. Again, high spatial variability of the susceptibility ranks was observed as related to the different combinations of factor maps. This made it difficult to delimit the ranks into susceptibility classes. Blahut et al. (2012) [25] studied the historical information on landslides and floods in the Tirano area for hazard estimation and definition of tentative risk scenarios. They developed a case study to exemplify the usefulness in their database of 489 records of damaging events (from the years 1600 to 2001), for realistic scenario generation, producing damage classification maps of territorial threats.

Recent analyses of the Tirano South database [16] focused on (1) credibility analysis of a fuzzy set modeled prediction pattern of landslide susceptibility and separation of well predicting from poorly predicting landslide occurrences [26] and (2) a generalized procedural strategy for comparisons of prediction patterns of active and dormant landslides by different models [27]. This contribution wants to expand that procedure and strategy attempting to interpret the uncertainties associated with target patterns and their consequences for understanding the prediction patterns generated by the application of those very same models.

## 4. Experimental Results

The Tirano South database, shown in Figure 2, was reanalyzed in order to expose the uncertainties associated with prediction patterns. The stepwise procedure for generating and assessing prediction patterns, proposed by Fabbri et al. (2017) [27], was also followed here, i.e., (1) use models to obtain prediction patterns from likelihood ratios, (2) cross-validate the patterns, (3) interpret the cross-validation results, and (4) obtain and compare the target and uncertainty patterns via equal-area ranks. Furthermore, step (5) was added, namely, analyze target, uncertainty, and 50% combination pattern relationships.

Table 1 shows the likelihood ratios for the ISPs computed for the Tirano South study area. For the ISP units and value ranges, only the ratios $\geq 2$ or $\approx 2$ are shown in the table. A ratio of 2 represents a normalized frequency in the presence of occurrences that is twice that in their absence. The arbitrary value of 2 was used as an empirical rule of thumb to separate the supportive ISP units or value ranges from the unsupportive ones of the proposition in (1). A ratio of 1 represents that the frequency in the presence of occurrences is the same as the one in their absence, i.e., null support of the proposition. Only the

maximum values of the supportive ratios and respective units and value ranges are shown in Table 1: two land-use classes (2.29 and 4.48); eight lithology units (5.19 to 5.52); three permeability classes (1.92, 2.06 and 1.91), (categorical ISPs); one range of aspect angles (maximum ratio 2.07); two curvature ranges (11.07 and 5.62); two elevation ranges (4.24 and 2.31); one internal relief range (5.97); and one slope angle range (4.26). All prediction patterns generated by the five models have been based on the supports provided by all the ratio values of which, for simplicity, only the most supportive are listed in Table 1.

**Table 1.** Subset is listed of categorical and continuous field ISPs in the Tirano South study area and their respective empirical likelihood ratio values. They will be used for predictions using as DSP the distribution of the set of landslides, **70at** with **ulpACDIS** as ISPs. Abbreviations for categorical ISPs are $\mathbf{u}_{1-23}$, land-use classes; $\mathbf{l}_{1-51}$, lithology units; and $\mathbf{p}_{1-8}$, permeability classes. For the continuous field ISPs, A, C, D, I, and S are aspect, curvature, digital elevation, internal relief, and slope, respectively. Values are bold if the ELR $\geq 2.00$. In Italics is the corresponding range of values, with the maximum ratio in brackets. In this reduced version, only ratios $\geq 2$ or $\approx 2$ are shown (table modified after Tables 1 and 2 in Fabbri et al., 2017 [27]).

| **Supporting Categorical Units or Classes Converted to ISPs for DSP 70at** | |
| --- | --- |
| Land use $\mathbf{u}_{1-23}$ | $\mathbf{u}_2$, Rock and scree vegetation; $\mathbf{u}_3$, Bedrock outcrops and surficial deposits. |
| Lithology $\mathbf{l}_{1-51}$ | $\mathbf{l}_6$, Sandstones; $\mathbf{l}_{20}$, Non-vegetated deposit; $\mathbf{l}_{26}$, Active non-vegetated scree slope; $\mathbf{l}_{40}$, Outcropping quartzites; $\mathbf{l}_{44}$, Outcropping ypo-abyssal rocks; $\mathbf{l}_{45}$, Intrusive rocks; $\mathbf{l}_{46}$, Outcropping intrusive rocks; $\mathbf{l}_{49}$, Serpentinites. |
| Permeability $\mathbf{p}_{1-8}$ | $\mathbf{p}_2$, Cohesive units with low permeability; $\mathbf{p}_3$, Cohesive units with very low permeability; $\mathbf{p}_4$, Non-cohesive units with high-medium permeability. |
| **Categorical ISPs with ELR values $\geq 2$ or $\approx 2$** | |
| Land use $\mathbf{u}_{1-23}$ <br> Lithology $\mathbf{l}_{1-51}$ <br> Permeability $\mathbf{p}_{1-8}$ | $\mathbf{u}_2 = \mathbf{2.29}$, $\mathbf{u}_3 = \mathbf{4.48}$; <br> $\mathbf{l}_6 = \mathbf{5.19}$, $\mathbf{l}_{20} = \mathbf{18.91}$, $\mathbf{l}_{26} = \mathbf{5.89}$, $\mathbf{l}_{40} = \mathbf{6.15}$, $\mathbf{l}_{44} = \mathbf{2.41}$, $\mathbf{l}_{45} = \mathbf{13.10}$, $\mathbf{l}_{46}\mathbf{l}_{46} = \mathbf{19.79}$, $\mathbf{l}_{49} = \mathbf{5.52}$; <br> $\mathbf{p}_2 = \mathbf{1.92}$, $\mathbf{p}_3 = \mathbf{2.06}$, $\mathbf{p}_4 = \mathbf{1.91}$. |
| **Continuous field ISPs with ELR values $\geq 2$** | |
| Aspect (0°-359°), **A** <br> Curvature ($-32$–+29), **C** <br> D. Elevation (350–2906m), **D** <br> Int. Relief (0–111 m, 3 × 3 pixels), **I** <br> Slope (0°–61°), **S** | $\geq 2$: *168–198*, (max **2.07**); <br> $\geq 2$: *$-24$–$-7$*, (max 11.07); *+7–+17*, (max **5.62**); <br> $\geq 2$: *1737–2104*, (max **4.24**); *2484–2629*, (max **2.31**); <br><br> $\geq 2$: *24–56*, (max **5.97**); <br> $\geq 2$: *37–57*, (max **4.26**) |

### 4.1. Use Different Mathematical Models to Obtain Prediction Patterns from Likelihood Ratios

The empirical likelihood ratios represent the spatial relationships within the database. They were obtained using the trigger zones distribution image of the 70 active landslides, **70at**, as DSP, and the ratio transformed images of the three categorical and the five continuous-field maps, **ulpACDIS**, as ISPs. Each model was applied to generate a different prediction pattern. The models integrated the ratios for each ISP. The patterns, displayed in Figure 3, were generated by the five mathematical models each requiring different assumptions and providing incompatible scores computed from identical inputs of likelihood ratios. To identify the patterns and their inputs in the computations, a sequence of short names was used as follows: MODEL_DSP_ISPs, as in **FZ_70at_ulpACDIS** to **BP_70at_ulpACDIS**. The one-letter ISP abbreviations were used in analyses with subsets of them.

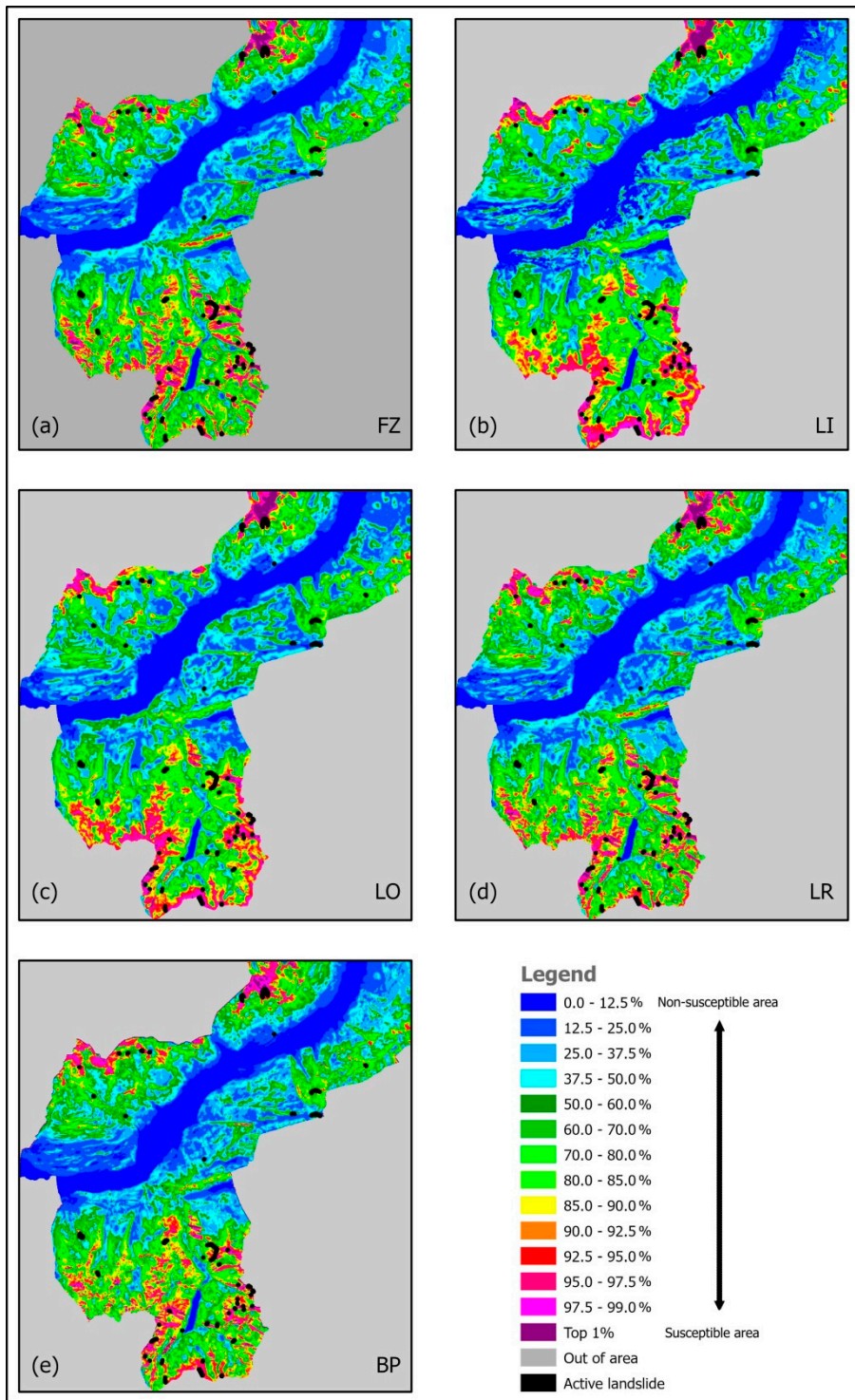

**Figure 3.** Prediction patterns using the different models: FZ, LI, LO, LR, and BP in (**a**–**e**), respectively, all using **70at_ulpACDIS**. Black patches are the oversized trigger zones of the 70 active landslides, **70at**. Colors in the legend indicate groups of ranks of % of the study area (SA) in ascending order.

A convenient way to interpret the scores resulting from the modeling is by converting them into 200 equal-area ranks after sequencing in descending order. Fixed recognizable groups of ranks are then associated with pseudo-colors, as shown in the legend of Figure 3. The 200 ranks are displayed in wider groups for lower ranks of lesser concern, and in successively narrower groups for higher ranks of greater concern, e.g., 12.5%, 10%, 5%, 2.5%, 1.5%, and top 1%, of the area of the SA.

The illustration shows the prediction patterns from the five models, all overlaid with the distribution as black polygons of the 70 active landslide occurrences. The patterns represent the likelihood of future landslide occurrences in the SA.

Comparing them, we can observe similarities among higher-ranking groups in the northern part of the study area but strong differences are found in the southern part. Wider patches with high values are in Figure 3b,c, from the LI and LO models, respectively. Discontinuous patches are in Figure 3a,c,d, for the FZ, LR, and BP models. Altogether, there is a similarity between Figure 3a,d,e), and some similarity between Figure 3b,c. Note that we have to compare the same classes, fixed groups of equal-area ranks, in zones of concern. They are the zones with relatively higher ranks but located far from the known occurrences. In particular, we can focus on the top 10% ranks (90–99% and top 1%) corresponding, for instance, to higher elevations (1740–2100 and 2480–2630 m a.s.l.), high slope angles (37–57°), and very low permeability, in addition to the particular types of land use and lithology (see Table 1). How "good predictors" are the prediction patterns? Which one is the best or preferable? For answering these questions, we will have to consider the prediction-rate curves in Figure 4, described in the next step. They will provide a measure of predictability of future landslide occurrences, i.e., in our case, the "next seven."

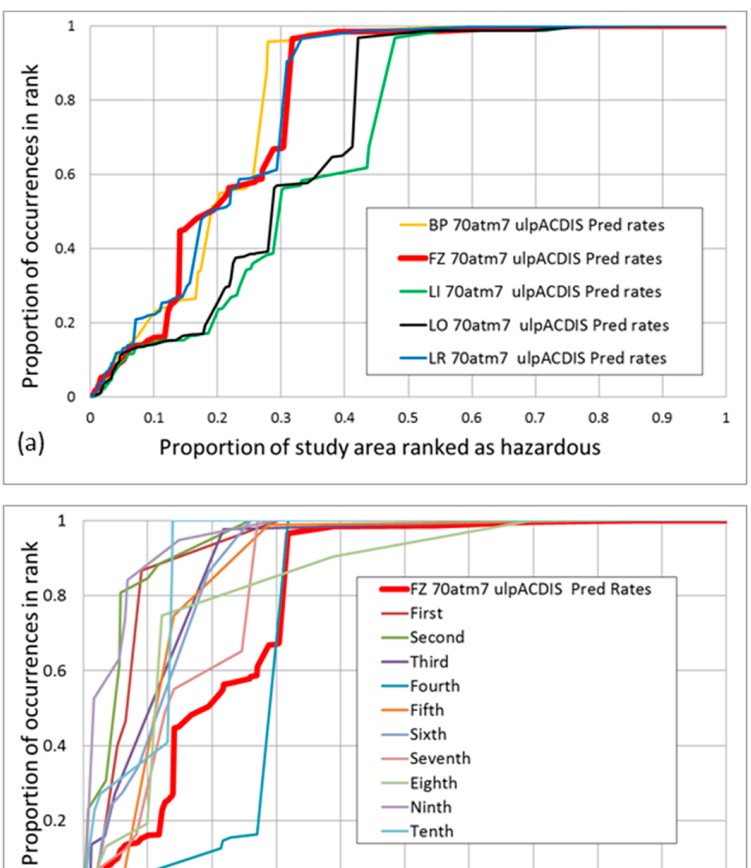

**Figure 4.** Prediction-rate curves are displayed generated by iterative cross validation using a sequential exclusion strategy of 7 out of 70 active landslides. (**a**) shows the aggregated iteration results for models FZ, LI, LO, LR, and BP, and (**b**) shows the individual curves for the 10 iterations of the FZ model and the aggregated curve as a thick red curve, corresponding to the red curve in (**a**).

### 4.2. Cross-Validate the Prediction Patterns

A cross-validation strategy was used to obtain and characterize the patterns as predictors. Iterative cross validation by the sequential exclusion of seven was tentatively selected out of the 70 active-landslide trigger zones in the DSP. It was computed to obtain the corresponding prediction-rate tables. The tables associate cumulative proportions of equal-area ranks of the prediction pattern with the corresponding cumulative proportions of validation occurrences. From the tables, cumulative curves are obtained, as shown in Figure 4. The illustration provides the prediction-rate curves from the iterative process using the five models. The horizontal axis shows the proportion of study area as cumulative equal-area ranks, each of 0.5% of the study area, i.e., ≈3230 pixels. The vertical axis shows the corresponding cumulative proportion of occurrence pixels in the cumulative class. The 697 occurrence pixels are distributed over the **70at** landslides so that the proportion of occurrence corresponds to the pixel numbers in each.

Note that the curves in Figure 4b represent proportions of seven occurrences, while the thick red curve represents the proportion of all 70 occurrences (curve **FZ_70atm7_ulpACDIS**, where m indicates minus).

The widespread distribution of the curves is indicative of the uncertainty affecting their aggregated thick red curve. What is predicted here is the likelihood of the "next" seven occurrences. In the iterations, they are assumed to be unknown. In all iterations, 63 occurrences are used to generate a prediction pattern that is then cross-validated as a predictor of the remaining seven occurrences. Note also that the proportions shown on the vertical axis are three orders of magnitude smaller than those on the horizontal axis (proportions of 697 pixels versus proportions of 646,091 pixels). As a reminder, the vertical axis was kept half the length of the horizontal axis.

We can observe that the prediction-rate curves are not particularly good. For instance, the thick red curve in Figure 4a shows that the top 10% ranks (0 to 0.1) contain about 14% of the occurrences for FZ, LI, and LO, and 22% for LR and BP; the top 20% ranks contain around 50% of the occurrences for FZ, LR, and BP, and about 26% for LI and LO; the top 30% ranks contain 96%, 91%, and 67% for BP, LR, and FZ, respectively, and about 57% for LI and LO.

These relative proportions represent the predictive capability of the modeling results: the proportion of predicted occurrences in the corresponding equal-area ranks. The shallow initial part of the curves is a sign of poor congruity of the setting of the landslide trigger areas used for cross validation. A good prediction pattern should provide an initially steep prediction-rate curve through cross validation in which most of the validation occurrences fall on higher ranks. The curves in Figure 4a show that the FZ, BP, and LR patterns of Figure 3 predict better than those for LI and LO. Figure 4b shows, in addition to the FZ prediction-rate curve identified with a thick red line (**FZ_70atm7_ulpACDIS**), the curves for each of the 10 iterations of the FZ model.

Note that a shallow initial curve is the one from the fourth iteration. Recall that in the iterations the cumulative proportions of occurrences in the diagram refers to the seven occurrences being cross-validated, while the proportions for the aggregated FZ curve refer to the 70 occurrences, validated into successive 10 groups of 7. What is being predicted here are the "next" 7 occurrences using the "previous" 63. This is the best approach when not having the time of the occurrences.

### 4.3. Interpret the Cross-Validated Results

The next step was to assemble sets of 10 prediction patterns to obtain via rank-based statistics the corresponding target and uncertainty patterns. The term "target" refers to what we are looking for as a validated prediction result. The term "uncertainty" refers to the stability of it. Indeed, the prediction pattern is the most informed prediction because it is using all information available, all the 70 active occurrences as DSP. However, we do not know its prediction capability. This is estimated by generating the target pattern via iterative cross validations using whichever strategy we can apply.

Figure 5 shows the uncertainty patterns obtained applying the five models to the database, the **70at** as DSP and **ulpACDIS** as ISPs. To obtain the target patterns, rank-based statistics was used to select for each pixel the corresponding median rank of the 10 prediction patterns generated by the iterative cross-validation process **70atm7**. The display of the target patterns is not provided here due to the extreme similarity with the corresponding prediction patterns in Figure 3 when using the same color legend. More informative are the uncertainty patterns in Figure 5, generated by ranking the ranges of ranks around the median ranks of the target pattern. The wider is the range the higher is the uncertainty of the corresponding target pattern (and consequently also of the prediction pattern).

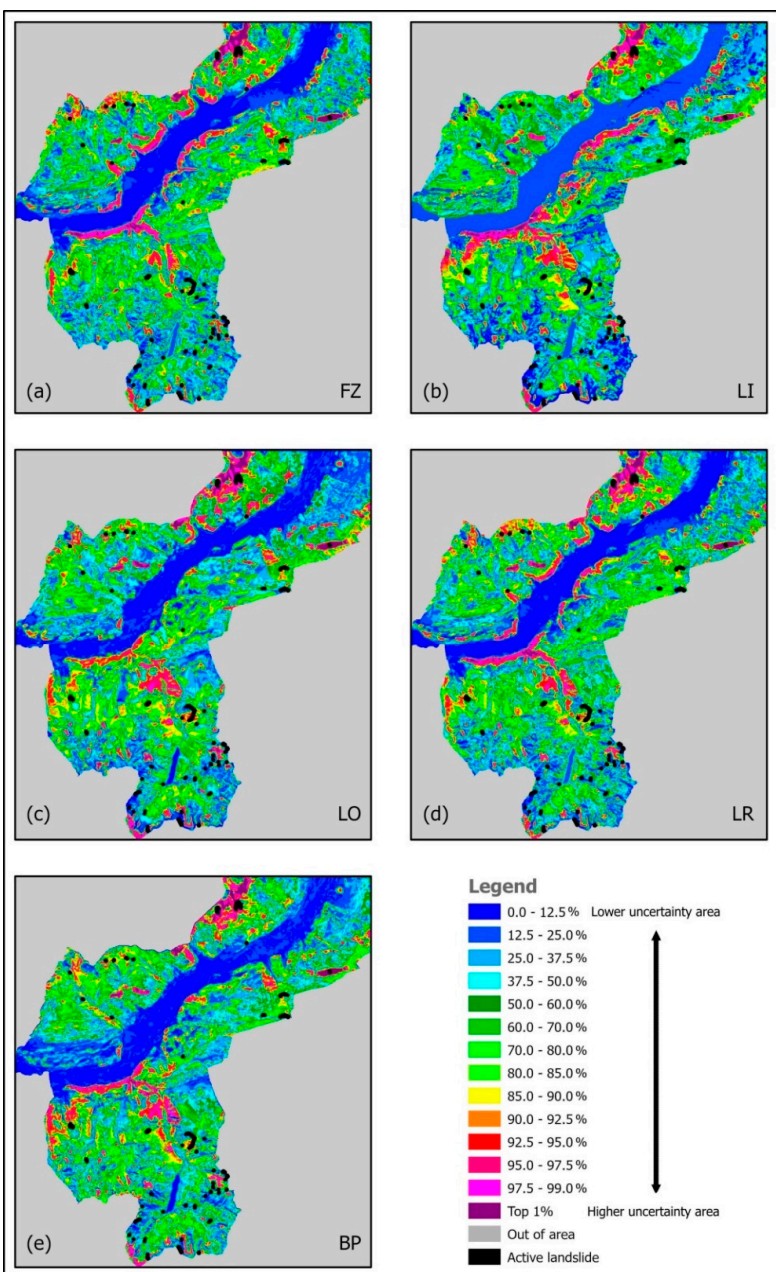

**Figure 5.** Uncertainty patterns, all using **70atm7_ulpACDIS** as cross-validation process, from the different models FZ, LI, LO, LR, and BP in (**a–e**), respectively. Black patches are the oversized trigger zones of the 70 active landslides. Colors in the legend indicate groups of ranks of % of SA in ascending order.

We can use the same color legend for the uncertainty ranks as for the prediction or target ranks. The uncertainty ranks relate to all target ranks from high to low. For uncertainty, however, the desirable ranks are the lower ones, indicating lower uncertainties. Obviously, if a high target rank corresponds to a high uncertainty rank, it is considered less credible than one corresponding to a low uncertainty rank. Figure 5 shows areas of high uncertainty, higher ranks, to the North in all the five uncertainty patterns. As to those along the valley edges, they are visible only in Figure 5a,b,d, for FZ, LI, and LR, respectively. Other areas of high uncertainty to the south are common to all the five patterns in the illustration.

At this point, it becomes important to study the relationships between target and corresponding uncertainty ranks with the 70 active landslide occurrences, **70at**, from the **70atm7_ulpACDIS** process of cross validation.

### 4.4. Obtain and Compare the Target and Uncertainty Patterns via Equal-Area Ranks

By cross-validating the target and uncertainty patterns with the 70 active landslides, **70atm7**, the relationship between target and uncertainty ranks were visualized, conveniently expressed in *1000 units (for instance, rank 900 corresponds to the top 10% equal-area rank). Figure 6 shows as an example the relationship between the target ranks, in descending order on the horizontal axis, and the respective uncertainty ranks, in ascending order on the vertical axis. It was obtained from the cross-validation process **FZ_70m7at_ulp_ACDIS.** In the illustration, the diagram shows that the 70 points are distributed so that the higher target ranks on the horizontal axis appear to correspond to relatively lower uncertainty ranks. It is worth noting that there are five encircled points corresponding to areas of relatively high uncertainty ranks and the lowest target ranks. They are occurrences that contribute to the shallow part of the prediction-rate curves in Figure 4. They could be considered outliers amongst the occurrences used as DSP.

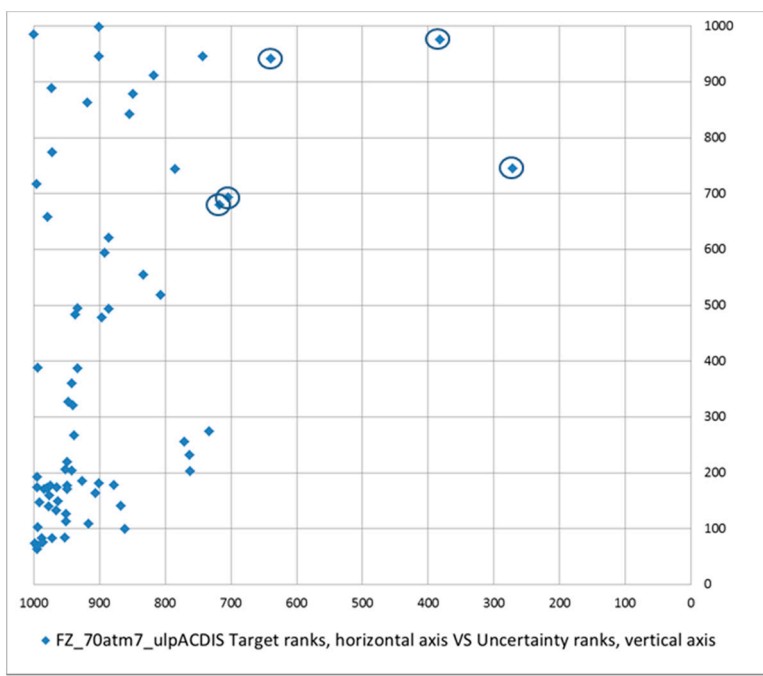

**Figure 6.** Target ranks in *1000 for the FZ model on the horizontal axis versus uncertainty ranks on the vertical axis, for the 70 active landslides, **70at**, in the Tirano South study area. The five encircled points have the lowest target ranks. See text for explanation.

Such a tendency for the patterns from the five models can also be visualized by sequencing the target ranks of the 70 occurrences in decreasing order, constructing histograms with pair of columns of target and corresponding uncertainty ranks. This is accomplished

in Figure 7 that shows in blue the target ranks and in red the corresponding uncertainty ranks for each occurrence. For all models, the histograms express a preferential distribution of higher uncertainty ranks for lower target ranks.

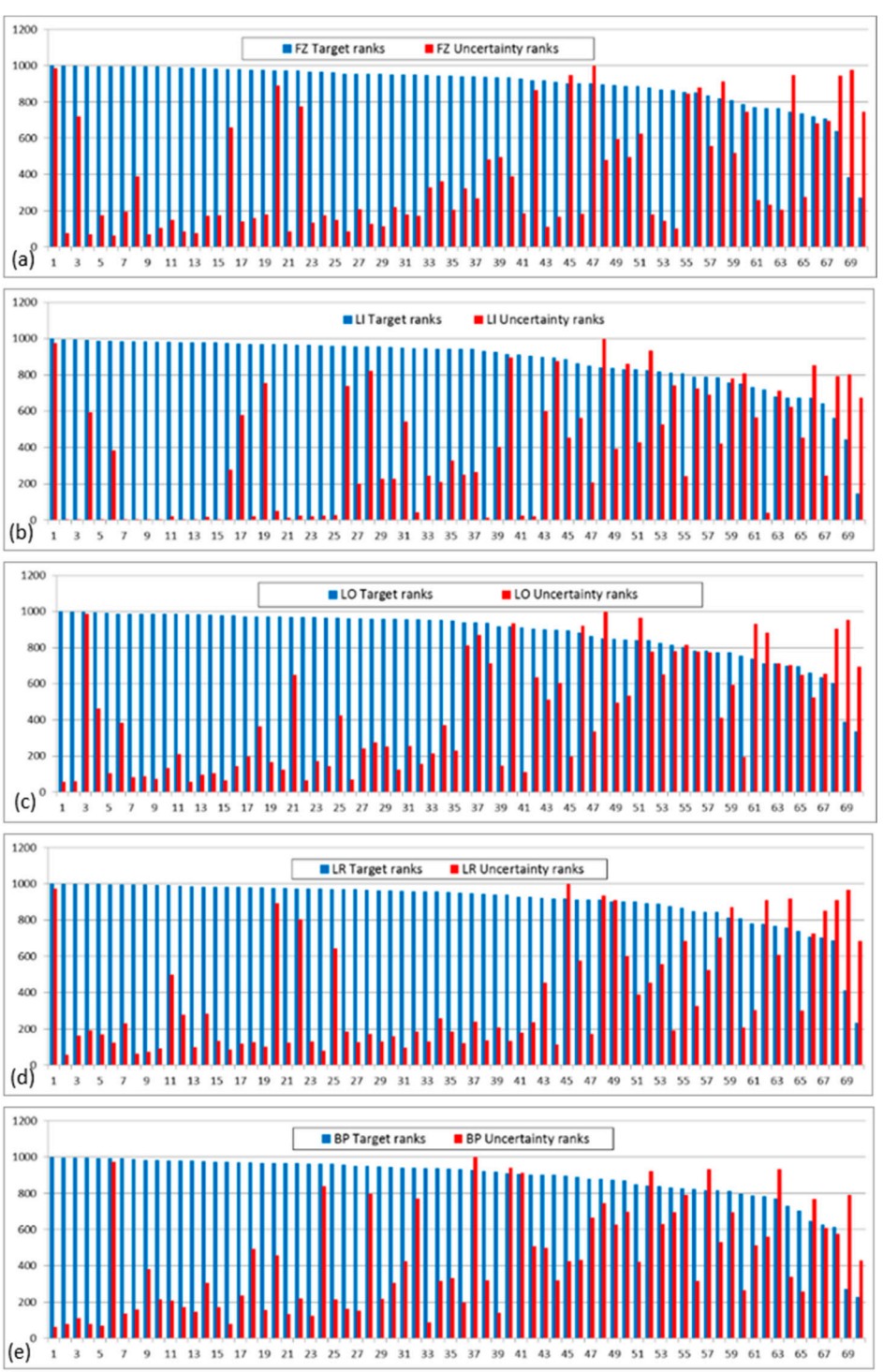

**Figure 7.** Histograms of decreasing target ranks (blue columns) and corresponding uncertainty ranks (red columns) for models FZ, LI, LO, LR, and BP from (**a**–**e**), respectively, using the cross-validation process **70atm7_ulpACDIS**.

We may wonder whether that tendency is visible also in the target patterns (or in the prediction patterns). This can be observed by generating combination patterns that relate uncertainty and target ranks by tentatively thresholding the uncertainty patterns. We have

generated the 50% combination patterns for each target pattern, as shown in Figure 8. A threshold value was arbitrarily set at the lower 50% uncertainty ranks in the study area to select the corresponding target ranks (or alternatively, we could have used prediction ranks from the respective prediction patterns).

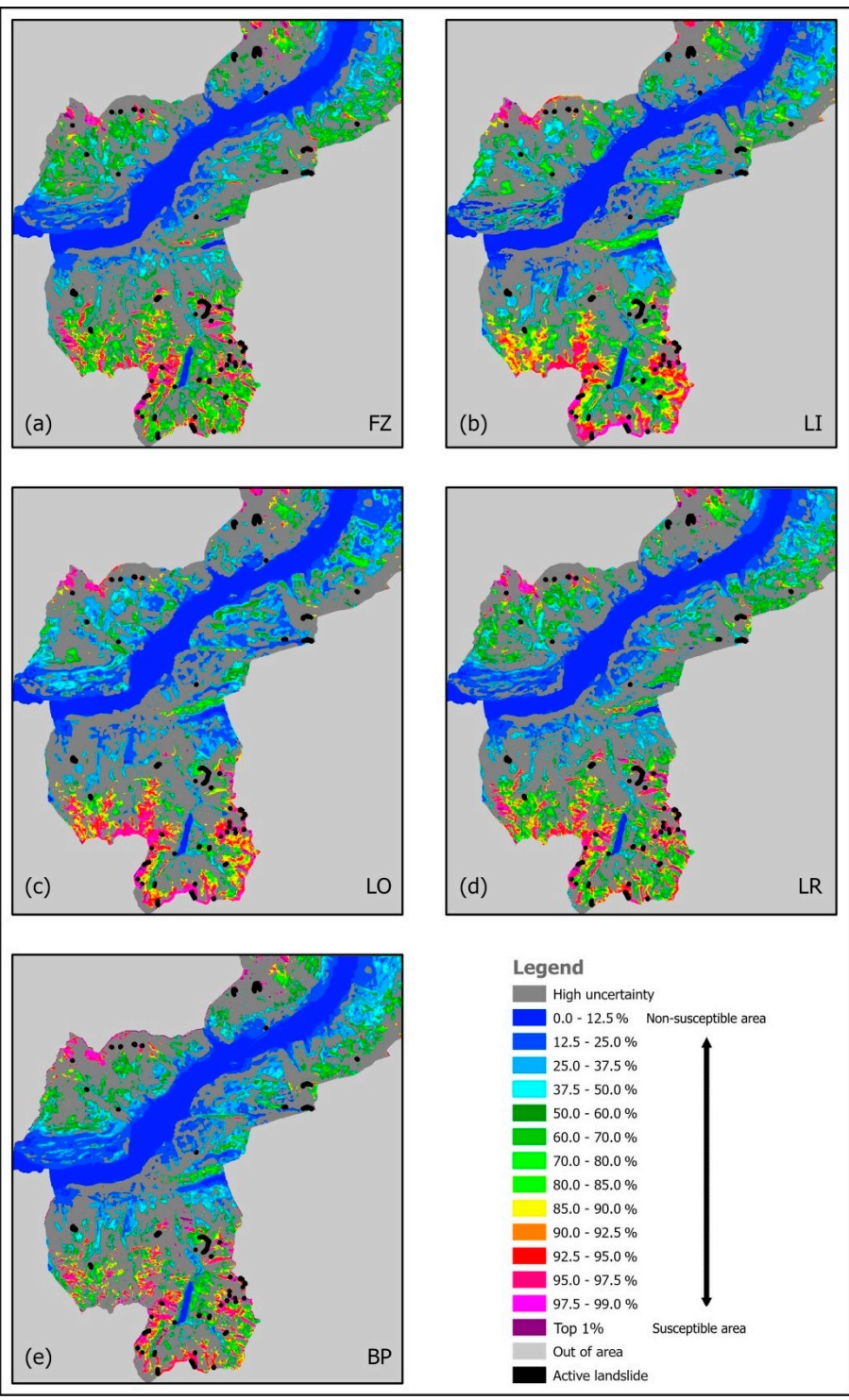

**Figure 8.** The 50% combination patterns, **70atm7**, using the different models FZ, LI, LO, LR, and BP in (**a–e**), respectively. Black patches are the oversized trigger zones of the 70 active landslides, **70at**. The dark-gray areas represent 50% of the study area with relatively higher uncertainty. The remaining 50% show the target pattern ranks for areas with lower uncertainty. Colors in the legend indicate groups of ranks of % of SA in ascending order.

By comparing the uncertainty pattern in Figure 5 with the prediction patterns in Figure 3, we can infer that the combination patterns in Figure 8 have retained high values in the southern part of the SA that shows low uncertainty (lower 50% uncertainty ranks). In the northeastern part, however, uncertainty is relatively high (upper 50% uncertainty ranks) so that the high target ranks are not visible in the combination patterns. What are then the characteristics of the 50% combination patterns? How is the uncertainty range threshold shaping them?

Let us consider some revealing details of the rank distribution in a small window of the 50% combination patterns, as shown in Figure 9. Seven landslide trigger areas are displayed as black contours in the illustration. Some are predicted as uncertain and fall on gray areas, in Figure 9b,c,e by LI, LO, and BP models. On the contrary, they have low uncertainty in Figure 9a,d by FZ and LR models. Topologically, similar dispersed patches are visible in Figure 9a,d,e, and more continuous ones in Figure 9b,c. The top 1% combination ranks, purple color, show the high variability of these clusters of pixels. These are the main characteristics of the patterns. They indicate weak robustness of the ranking and the uncertainty at the occurrence location.

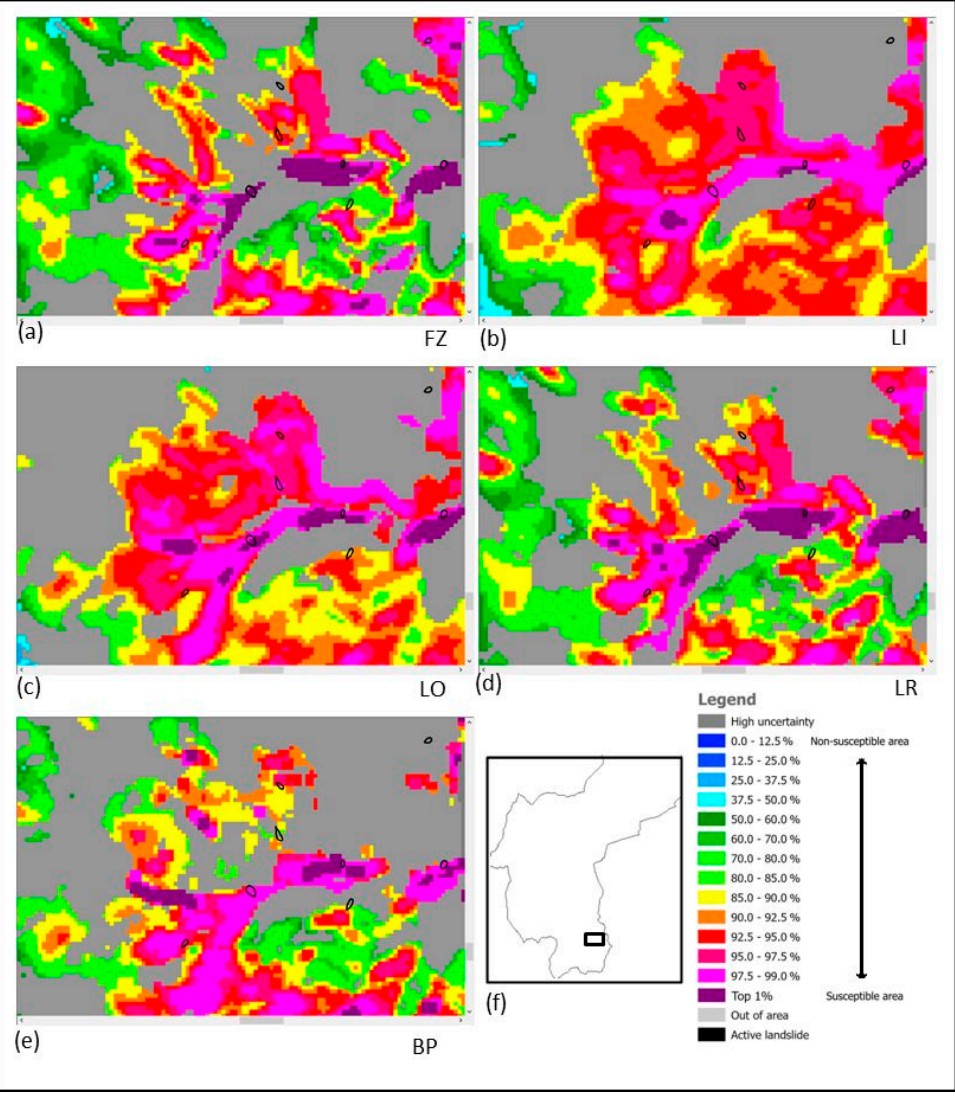

**Figure 9.** A particular small subarea is shown of 50% combination patterns with overlaid boundaries of the seven active landslides located in it. In (**a**,**b**) are the patterns from models FZ and LI; in (**c**–**e**) from models LO, LR and BP, respectively. In (**f**) is the location of the subarea. Colors in the legend indicate groups of ranks of % of SA in ascending order.

Next, we can derive a more general characterization of an entire 50% combination pattern by looking at low-uncertainty proportions or pixel numbers in the individual 200 ranks.

### 4.5. Analyze the Target, Uncertainty, and 50% Combination Pattern Relationships

Increasing uncertainty ranks with decreasing target ranks, as we have seen with the **70at** landslide trigger areas via the **70atm7** process in Figures 6 and 7, are also found in the target patterns. We can visualize their suspected higher uncertainty (hinted by the curves in Figure 4b) for lower prediction ranks. This is carried out by generating plots similar to the one in Figure 6 but with 646,091 points corresponding to the pixels in the SA, using pairs of target and uncertainty patterns. Instead, the following simpler visualization procedure was followed.

We have generated 200 equal-area ranks out of the prediction, target, and uncertainty ranks, each corresponding to 0.5% of the study area ($\approx$3230 pixels). Being concerned mostly with the highest ranks, e.g., the higher 80%, we have displayed the numbers of pixels in each combination pattern that corresponded with the 50% lower uncertainty ranks, leaving out those with higher uncertainty ranks. We observed, therefore, the decrease in pixel number for each rank due to the elimination of higher uncertainty ranking pixels and detected which ranks were consequently more uncertain. These were the intermediate ranks. Figure 10 compares this decrease in the 50% combination patterns from the five models with the respective target ranks, i.e., the straight lines. In all cases, the intermediate ranks show higher uncertainty, i.e., higher "loss" of target pixels in the combination ranks, between rank 160th and rank 80th. The curves in Figure 10 have strongly variable configurations with one or two concavities and different higher pixel numbers in the vicinity of the 200th and the 40th ranks. Figure 10f contrasts the curves from models FZ and LO. Note the absence of a drop in pixel number for the top ranks in Figure 10e. It shows a sharper pick, as to be expected from the yellow prediction-rate curve from model BP in Figure 4a that appears to be less sensitive to the suspected outlier occurrences.

We may wonder whether the curves indicate some form of database or modeling signature. The curves in Figure 10 show how the combination patterns reach high uncertainties at intermediate ranks. The relative uncertainties increase from initial lows to reach one or two maximum values between rank 190th and rank 60th. Since this is observed for all the patterns from the five models, it appears as a database property. Observe the prediction-rate curves in Figure 4a and consider the curves in Figure 10 showing the number of target ranks corresponding to lower uncertainties close to the two extreme ranks displayed. Would we want to select the top 10% combination ranks, from 200th to 180th, as the acceptable part? Or alternatively, would we prefer the top 20% as more significant? The properties of uncertainty and combination patterns are still unknown and remain a research challenge.

### 4.6. Considerations on Prediction Patterns as Maps

What resembles a map, such as the display of a prediction pattern, is not necessarily a map but more a representation of information extraction through normalizations, conversions, assumptions, and integrations. At present, with what we know about a study area, we have to be satisfied by the higher part of the prediction-rate curve and by the corresponding combination pattern, perhaps fine-tuning the process. Otherwise, additional information must be used to choose a satisfactory part of the prediction pattern. Regarding the prediction patterns, would we want to discover instances of unmapped landslide trigger zones in the higher 10% ranked areas away from the known ones? Is the 50% combination pattern helpful in mapping more of what we know or that we do not know? Should we wait for the next seven landslides to see where they will appear? Where should we concentrate on more detailed mapping? How much of the study area should we consider unfit for particular land uses or developments? Would we opt for using cost/benefit or using safety criteria? Providing answers for decision making and

subsequent risk analysis is the logical function of the prediction patterns of landslide susceptibility, as we have discussed.

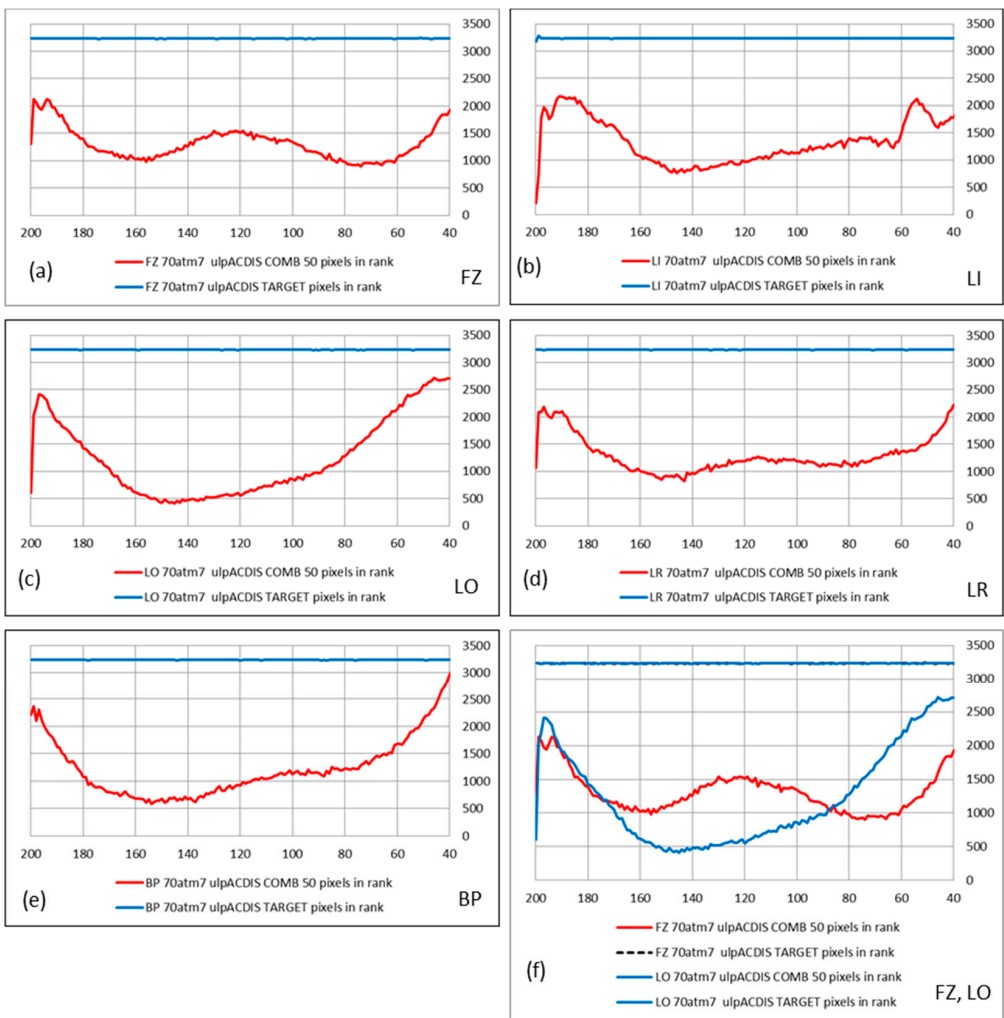

**Figure 10.** Comparison of pixel numbers in equal-area target ranks with the pixel numbers in the corresponding low uncertainty target ranks of the 50% combination pattern. Models FZ, LI, LO, LR, and BP are in (**a**–**e**). In (**f**) is the overlay of curves for models FZ and LO. The target equal-area ranks contain about 3230 pixels. Only the higher 160 ranks are shown here.

## 5. Concluding Remarks

Favorability function modeling was applied to the Tirano South database through five different models of spatial relationships. All the prediction patterns obtained were represented as relative ranks, including their derived target and uncertainty patterns. The 50% lower uncertainty ranks were tentatively used to extract the corresponding target ranks as 50% combination patterns. In them, for all models, the proportion of less uncertain pixels in the rank represents the level of confidence in the prediction. Possibly, the top 10% combination pixels are a significant part of the prediction pattern. We are trying to find answers to questions such as the following: What are the consequences of higher uncertainty for intermediate ranks in combination patterns? How are these diagrams providing a measure of the quality of the prediction? Are they characteristic of the modeling because of the model or the data? How much of the prediction pattern is reliable? How to evaluate the higher ranks in the prediction patterns? Should we select the top 10% or 20% combination ranks? What would our cost/benefit considerations and choices be? All we have generated from the modeling are "relative" integrated equal-area ranks that must be

interpreted as susceptibility to land-sliding. The model preference needs to be a function of the interpretation of prediction patterns and their representation. In our case, FZ, BP, and LR are equally satisfactory but less so for LI and LO, as to the pattern of predictive performance. However, this may be just one of the criteria that might be used.

A five-step procedure is proposed for modeling prediction and uncertainty. The uncertainty ranks are considered important to properly select the susceptible part of the study area, i.e., susceptible to the "next seven" future" landslide occurrences. While various other strategies have also been applied for iterative cross validation to obtain slightly different prediction patterns, the sequential exclusion of seven occurrences appears to do justice to the properties of the database. The procedure is proposed as critical in spatial prediction modeling. Independently of the models used, a necessary research issue is in the interpretation of the uncertainty associated with prediction patterns.

We have described our analysis and modeling results to indicate a way to predict and the assumptions implied. Obviously, we suspect that our experiments on this particular database have a more general significance beyond the specific study area or the five mathematical models used. These considerations, we hope, will be useful to researchers and users of susceptibility maps. This contribution does not provide a solution but poses questions whose answers point at possible solutions.

**Author Contributions:** Investigation, A.G.F. and A.P. All authors have read and agreed to the published version of the manuscript.

**Funding:** This contribution was initially and partly supported by the European Commission Project "Mountain Risks: from Prediction to Management and Governance" (MRTN-CT-2006-035978, 2007–2010), Mountainrisk (2007).

**Acknowledgments:** The authors are grateful to four anonymous reviewers and to Jan Blahut (The Czech Academy of Sciences) for helping to improve this manuscript.

**Conflicts of Interest:** The authors declare no conflict of interest.

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
