# Peer review of "Spatial Uncertainty of Target Patterns Generated by Different Prediction Models of Landslide Susceptibility"

_applsci, doi:10.3390/app11083341_

Round 1

Reviewer 1 Report

The authors present five methods to discuss the suitability for landslide susceptibility. However, there are several major issues need to improve.

  1. The five methods have different trainings or decision equations not shown in this article.
  2. The uncertainty factor is not clear supporting “assumption” or “prediction”.
  3. The conclusions don’t give extremely strong direction to use these five methods, which should provide more specific conditions or criteria.

Author Response

We are grateful for the reviewer’s remarks that are helpful in improving our manuscript.

  1. The five methods have different trainings or decision equations not shown in this article.

In reality one methodology is being provided by the article: the “favourability function” framework for spatial prediction modelling. The five mathematical models have been used to show that the framework, independently of the models, represents a strategy that leads to new results and information: target patterns and uncertainty patterns. This is so even if the five mathematical models imply different assumptions, normalizations, value ranges of the resulting scores and combination rules for integration. The scores become relatively comparable as equal area ranks. The particulars of the mathematical models are not of interest and are well known. We have reiterated this in lines 18-186. They are presented and discussed in the referenced contributions. We did not want to shift the focus of the paper. All changes are in bold red in the attached version of the edited manuscript.

  1. The uncertainty factor is not clear supporting “assumption” or “prediction”.

We are not using an “uncertainty factor” for supporting an “assumption” or a “prediction”. Given a proposition, i.e. a mathematical formulation to be supported as true, we are using spatial relationships in the database as support. Such relationships are integrated into equal-area ranked prediction patterns. To assess the predictive functionality of the ranks, the patterns are “disturbed” (by iterative cross-validation) to obtain target patterns. These new patterns, also equal-area ranked, enable estimating the uncertainty of the ranks. Target patterns are what we would like to have as prediction results. Of course, the higher the uncertainty the weaker becomes the rank in the target pattern. We would not know this from the prediction pattern without generating the target pattern. In the example of our applications the target pattern is generated by calculating the median rank of the sequence of ten prediction patterns obtained by cross-validation. The uncertainty pattern is obtained by ranking the range of ranks around the median in that sequence. The assumptions in the favourability modelling framework are prerequisite for justifying the significance and limitations implied in the modelling. We have tried to make it clearer in the text.

  1. The conclusions don’t give extremely strong direction to use these five methods which should provide more specific conditions or criteria.

Now that we have shown, in some applications and databases, what the target and uncertainty patterns look like and their hidden properties (not well known so far), more research is needed to interpret the relative significance and uncertainty. As to the results of the modelling, the  prediction patterns, either one is satisfied by the higher prediction-rate curve and combination pattern (and selects a convenient high ranking area in the pattern as susceptible area) or other observations have to be found to choose a satisfactory prediction pattern. We have just provided a methodology and one example of strategy of analysis that revealed the unknown. Clarifications in the text have been introduced (see Sub-section 4.5 and in Section 5, the Concluding remarks) to make sure that the issues raised by the reviewer are satisfactorily discussed including the strengthening of the direction for using the results of the analyses. Of necessity much has to be left up to the potential user or decision-maker’s task or scope.

Reviewer 2 Report

This study used five different spatial relationship models, the convenience function modeling was applied to the Tirano South database. A five step procedure was proposed for modelling prediction and uncertainty. It was concluded that uncertainty assessment is an ideal and necessary work to make spatial predictive modeling worthy of practice. This manuscript was almost complete, and there are only a few problems needed to be modified.

  1. In introduction, there are no citations in this section. It should be modified to cite enough literature to support your content.
  2. In figure 3, 5,6, the Legend should be modified to introduce the color meaning.
  3. There are a lot of initialism in the manuscript. I suggest that author could give a table to refer to the meaning.

Author Response

Thank you for helping us to improve the manuscript. Indeed your observations have identified some weak points.

  1. In introduction, there are no citations in this section. It should be modified to cite enough literature to support your content.

We have added four references [1-4] in support of the statements made in the introductions. All changes are in bold red in the attached version of the edited manuscript.

  1. In figure 3, 5, 6, the legend should be modified to introduce the color meaning.

We have improved the legend as suggested and added further explanations in the text and in the captions of Figure 3, 5, 8 and also Figure 9.

  1. There are a lot of initialisms in the manuscript. I suggest that authors could give a table to refer to the meaning.

Section 2, now entitled “Predictive methodology, Terminology and Analytical Procedures” develops “favourability function” terminology, gradually defining concepts going from the original spatial input data, via the proposition, to assumptions, spatial characterization, integration, interpretation of relative ranks and their visual rendering. All analytical steps in the process of “favourability modelling” require unavoidable arbitrary choices of parameters or units or representations. In each application such choices would depend on the characteristics of the database and may need to be adjusted to it, even if the general approach is to be the same. For this reason we feel that a table of definitions would not help to comprehend the relative significance of the modelling results.

Reviewer 3 Report

It's OK.

Author Response

  1. Description of methods can be improved, otherwise it is OK.

The methodology of “favourability function” modelling has been described. No details are provided on the five mathematical models in the application. These are well known and have been dealt with in the contributions referred to in the text. The methodology proposed is independent of the mathematical models. We have made it clearer in lines 183-186. All changes are in bold red in the attached version of the edited manuscript.

Reviewer 4 Report

I approach this manuscript as a user of susceptibility maps rather than as a researcher actively involved in the technical preparation of such maps.

I do think the work tackles an important but underestimated issue. On the other hand, I found the manuscript excessively technical and hard to follow.

The abstract, in particular, should be directed to a more general public, especially considering the fact that the work was submitted to "Applied sciences", a multi-purpose journal, and not to a technical venue of the field, such as "Remote sensing" or journals of other editors, e.g., Springer's Stochastic Environmental Research and Risk Assessment. See, e.g., the incipit "This contribution juxtaposes the relative uncertainties associated with target patterns of landslide susceptibility". Why "juxtaposes"? Perhaps "compare", "discusses"? What are the "relative uncertainties" in this context? What are "target patterns"? I am just afraid that the work would be only read by a small niche of experts evan though it has a potential larger importance.

Author Response

The authors are grateful for the reviewer’s remarks that identify points in our manuscript that need clarification or reassurance.

  1. I approach this manuscript as a user of susceptibility maps rather than as a researcher actively involved in technical preparation of such maps.

We think that it important that not only researchers but also users of susceptibility maps become aware of how spatial prediction maps have to be interpreted because of what they do represent. Portraying likelihoods of future occurrences landslides, i.e. landslide susceptibility, is not a simple process, so that awareness of what is represented and how it can be interpreted is of critical importance. We have added Sub-section 4.5 to discuss this point and touched on it at the end of the concluding remarks. All changes are in bold red in the attached version of the edited manuscript.

  1. I do think the work tackles an important but underestimated issue. On the other hand, I found the manuscript excessively technical and hard to follow.

The problem of spatial prediction modelling is complex and requires unavoidable assumptions. Such assumptions are often omitted or not recognized even in specialized literature. The mathematical and statistical concepts are also complex but well known and established mathematical models for spatial modelling are available (we make reference to five models used in our application example) and new ones are continuously being proposed. What we think is more important than the mathematical models is to respect the information content of the database and to extract its suitability to prediction of susceptibility to land sliding. It is unfortunate that our description appears hard to follow! It is also hard to model correctly and to communicate the results to different potential users. Here our scope is to point at a strategy of analysis that brings out what is hidden in most of the modelling applications. The technicality has the purpose of helping in the understanding of the “relative” meaning of the results, not so much the preparation but the meaning of the relative (equal-area) ranks with respect to the initial spatial input.

  1. The abstract, in particular, should be directed to a more general public, especially considering the fact that the work was submitted to "Applied sciences", a multi-purpose journal, and not to a technical venue of the field, such as "Remote sensing" or journals of other editors, e.g., Springer's Stochastic Environmental Research and Risk Assessment. See, e.g., the incipit "This contribution juxtaposes the relative uncertainties associated with target patterns of landslide susceptibility". Why "juxtaposes"? Perhaps "compare", "discusses"? What are the "relative uncertainties" in this context? What are "target patterns"? I am just afraid that the work would be only read by a small niche of experts even though it has a potential larger importance.

The abstract has been modified and simplified, also replacing “juxtaposes” with “exposes.

Relative uncertainties are defined as ranges around a mean value (a median in our case) of the pixels in the target pattern. The wider is the range, the greater is the uncertainty. The target pattern is generated by calculating the median of all the prediction patterns obtained by iterative cross-validation: a process that, in our instance, sequentially excludes a few landslides to obtain a prediction and systematically generates a desired number of slightly different prediction patterns (whose statistics allows estimating uncertainty). It is what we want to have if it represents the prediction of future occurrences while allowing us to estimate the uncertainty associated to it: the uncertainty of the ranks of the target patterns and as a consequence associated to the prediction pattern.

We wanted to have a representation of the future susceptible areas that not only classifies levels of susceptibility but also the uncertainty associated with those levels. This is, hopefully, the target pattern (should the database be good enough). It tells what we want to (or can) know of the prediction pattern. To obtain the target pattern we have been literally “torturing” the prediction pattern by iterative cross-validation. Why talking of “patterns”? Because the modelled scores are by themselves “uninterpretable”: they have to be transformed into relative ranks to be visualized and interpreted. What looks like a map is not necessarily a map but a representation for information extraction to conveniently generate maps (see the new Sub-section 4.5). We have opted for equal-area ranks, each of 0.5% of the study area. In our applications we have selected the median rank to obtain a target pattern and the relative ranking of the ranges around the median values to obtain the uncertainty. This could be objected to. We could have used other statistics but we found the median and range to be the most robust. In the paper we have described how all this was done to indicate a way to predict and the assumptions required. Obviously, we suspect that our experiments on this particular database have a more general significance beyond the particular study area or the specific mathematical models used. These considerations, we hope, should be useful not only for researchers but also for users of susceptibility maps (see the added text in the Concluding remarks).

Round 2

Reviewer 1 Report

The authors have made great improvement to this paper. I have no further questions.

Author Response

The authors thanks the reviewers for their positive comments 

Reviewer 2 Report

This manuscript has been modified and complete. I believe that the article can be published as a revised version.

Author Response

The author thanks the reviewers for their positive comments